# Association of Hypoglycemia with Biomarkers of Oxidative Stress and Antioxidants: An Observational Study

**DOI:** 10.3390/healthcare10081509

**Published:** 2022-08-10

**Authors:** Eleftheria Papachristoforou, Aikaterini Kountouri, Eirini Maratou, Dimitris Kouretas, Zoi Skaperda, Maria Tsoumani, Panagiotis Efentakis, Ignatios Ikonomidis, Vaia Lambadiari, Konstantinos Makrilakis

**Affiliations:** 1First Department of Propaedeutic Internal Medicine, Laiko General Hospital, Medical School, National and Kapodistrian University of Athens, 11527 Athens, Greece; 2Second Department of Internal Medicine, Research Unit and Diabetes Centre, Attikon Hospital, Medical School, National and Kapodistrian University of Athens, 12462 Athens, Greece; 3Laboratory of Clinical Biochemistry, Attikon University Hospital, 12462 Athens, Greece; 4Department of Biochemistry and Biotechnology, University of Thessaly, 41500 Larissa, Greece; 5Laboratory of Pharmacology, Faculty of Pharmacy, National and Kapodistrian University of Athens, 15771 Athens, Greece; 62nd Department of Cardiology, Attikon Hospital, Medical School, National and Kapodistrian University of Athens, 12462 Athens, Greece

**Keywords:** hypoglycemia, oxidative stress, routine clinical practice

## Abstract

Hypoglycemia has been associated with complications from the vasculature. The contributing effects of oxidative stress (OS) on these actions have not been sufficiently studied, especially in daily, routine clinical practice. We examined the association of hypoglycemia encountered in daily clinical practice with biomarkers of OS and endogenous antioxidant activity in persons with diabetes [type 1 (T1D) or type 2 (T2D)], as well as individuals without diabetes, with a history of hypoglycemia. Several biomarkers of OS (MDA, ADMA, ox-LDL, 3-NT, protein carbonyls, 4-HNE, TBARS) and antioxidant capacity (TAC, superoxide scavenging capacity, hydroxyl radical scavenging capacity, reducing power, ABTS) were measured. Blood was drawn at the time of hypoglycemia detection and under euglycemic conditions on a different day. A total of 31 participants (mean age [±SD] 52.2 ± 21.1 years, 45.2% males) were included in the study. There were 14 (45.2%) persons with T2D, 12 (38.7%) with T1D, and 5 (16.1%) without diabetes. We found no differences in the examined biomarkers. Only TBARS, a biomarker of lipid peroxidation, showed lower values during hypoglycemia (*p* = 0.005). This finding needs confirmation in more extensive studies, given that MDA, another biomarker of lipid peroxidation, was not affected. Our study suggests that hypoglycemia encountered in daily clinical practice does not affect OS.

## 1. Introduction

Hypoglycemia represents the commonest acute complication of antidiabetic treatment, especially when medications like insulin or sulfonylureas are used [1,2], and may also be associated with increased cardiovascular risk [3,4,5]. It induces inflammatory, prothrombotic, antifibrinolytic, and arrhythmogenic responses [6], and has been associated with vascular complications [7]. Hypoglycemic episodes stimulate the sympathoadrenal system and increase the plasma levels of both epinephrine and norepinephrine, resulting in symptoms of sweating, trembling, anxiety, hunger, and nervousness [8], leading to overwhelming hemodynamic and hemorheological effects [9,10,11,12,13].

Oxidative stress (OS) has been implicated in the development of diabetes complications by causing endothelial dysfunction and inflammation [14], and the possible link of hypoglycemia with OS has been investigated as a causative factor for these complications [6,11,15]. However, the exact mechanisms are not well elucidated, especially in daily clinical practice. This study aimed to examine the association of hypoglycemia occurring in daily, routine clinical practice with oxidative stress and endogenous antioxidant activity biomarkers.

## 2. Materials and Methods

### 2.1. Participants

We examined persons with type 1 (T1D) or type 2 (T2D) diabetes who experienced spontaneous hypoglycemia during a routine clinical visit in the outpatient Diabetes Clinics of two major university hospitals, as well as non-diabetic individuals with a history of fasting or postprandial hypoglycemic episodes which were evaluated for the cause (possible insulinoma) with a prolonged fasting test. Hypoglycemia was considered present when symptoms consistent with hypoglycemia occurred [7], and finger-stick blood glucose levels < 3.33 mmol/L (60 mg/dL) (confirmed by venous blood) were detected.

Exclusion criteria included any condition that could adversely affect OS biomarkers, i.e., acute kidney injury, infection/sepsis, acute liver injury, recent (last three months) acute cardiovascular events, new-onset neoplasia, etc.

Regarding individuals without diabetes, there were four with a history of fasting hypoglycemia and one with postprandial hypoglycemic episodes, for whom there was suspicion of insulinoma. Subjects who experienced fasting hypoglycemic episodes were admitted to the hospital and underwent a 72-h fasting test. Blood glucose levels were measured every 4–6 h. The fast was discontinued when symptoms developed in the presence of hypoglycemia or when blood glucose levels were <3.89 mmol/L (70 mg/dL) without symptoms or signs, but Whipple’s triad had been documented. The non-diabetic subject with a history of postprandial hypoglycemic episodes underwent a prolonged oral glucose tolerance test (with a 75-g oral glucose load) over 4 h. Blood was drawn at half-hourly intervals or when symptoms developed in the presence of hypoglycemia. In all participants, there was a second blood sampling on another day, under stable euglycemic conditions (blood glucose levels 5.00–7.22 mmol/L [90–130 mg/dL]), for measurement of OS and antioxidant biomarkers. Citrated plasma (EDTA tubes) and serum samples were collected at the predetermined time points. They were separated immediately and frozen at −80 °C until analysis for the soluble biomarkers.

All subjects were asked to sign an informed consent. The protocol was aligned with the principles of the Declaration of Helsinki [16] and was approved by the hospital’s Ethics Committee.

### 2.2. Measurement of Redox Biomarkers

Oxidative stress biomarkers and antioxidant capacity that were determined at the time of hypoglycemia and subsequently during euglycemia were the following:

Oxidative stress biomarkers: Asymmetric dimethylarginine (ADMA, competitive ELISA assay, BioVision, San Francisco, CA, USA); oxidized LDL (ox-LDL, ELISA assay, Abcam, San Francisco, CA, USA); 3-nitrotyrosine (3-NT, ELISA assay, BioVision, San Francisco, CA, USA), were measured as per the manufacturer’s instructions. Malondialdehyde (MDA) and 4-Hydroxynonenal (HNE) were determined spectrophotometrically at 586 nm and expressed as μM using the Oxford Biomedical Research Colorimetric Assay for lipid peroxidation, as previously described [17]. A spectrophotometric measurement of 2,4-dinitrophenylhydrazine (DNPH) derivatives of protein carbonyls was used to quantify protein carbonyl (PC) content [18] and thiobarbituric acid reactive substances (TBARS).

Antioxidant capacity biomarkers: total antioxidant capacity (TAC, mmol DPPH/L), ABTS [2,2′-azino-bis(3-ethylbenzothiazoline-6-sulfonate) radical cation], superoxide scavenging capacity (mmol NBT/L), hydroxyl radical scavenging capacity (mmol deoxyribose/mL), reducing power (Ferric iron-reducing antioxidant power, FRAP assay).

Determination assays for these biomarkers have been previously described [19,20].

### 2.3. Statistical Analysis

Continuous variables are presented as mean ± one-standard deviation, while the qualitative variables are presented as absolute and relative frequencies (%). The normal distribution of variables was tested with the Shapiro–Wilk test. Comparisons between two normally distributed continuous variables were performed with the calculation of the paired-samples *t*-test, whereas the Wilcoxon signed-ranks test was used for non-parametric variables. Associations between categorical variables were tested with the use of contingency tables and the calculation of the Chi-squared test. Pearson’s correlation coefficient (r) or Spearman’s Rho coefficient (for non-normal distributions) were used for the evaluation of statistical correlations between variables. For comparisons of ≥3 variables, one-way analysis of variance (ANOVA) (for normally distributed variables) or the Kruskal–Wallis test (for non-normally distributed variables) was used. Data were analyzed using the Statistical Package SPSS, version 21.0 (SPSS Inc., Chicago, IL, USA).

## 3. Results

A total of 31 participants (mean age [±SD] 52.2 ± 21.1 years, 45.2% males) were examined, 14 (45.2%) with T2D (60.8 ± 21.3 years), 12 (38.7%) with T1D (44.3 ± 17.8 years) and 5 (16.1%) were non-diabetic individuals (47.2 ± 22.2 years). Concerning diabetic persons, the mean disease duration was 19.9 ± 11.4 years, and HbA1c was 7.5 ± 1.2%. All patients with diabetes who developed hypoglycemia happened to be treated with insulin, together with appropriate lifestyle advice (and other medications as appropriate for T2D). None of the persons that participated in the study were being treated with sulfonylureas.

Table 1 depicts the study participants’ demographic, clinical, and laboratory characteristics. As expected, people with T2D were more obese than the other two groups (*p* = 0.028). People with type 1 diabetes had a longer duration of diabetes as compared with type 2 diabetic individuals.

The comparison of oxidative stress and antioxidant capacity biomarkers between hypoglycemia and euglycemia states is shown in Table 2. Levels of TBARS, a marker of lipid peroxidation, decreased significantly in hypoglycemia (6.9 ± 2.7 μmol/L) as compared with euglycemia (9.2 ± 3.7 μmol/L) (*p* = 0.005 by Wilcoxon signed-ranks test) (Figure 1). All other biomarkers did not show significant differences between hypoglycemia and euglycemia.

Simple correlation analyses showed that TBARS levels during euglycemia were negatively correlated with age (r = −0.364, *p* = 0.044). In addition, TBARS levels during euglycemia were significantly higher in males as compared with female subjects (10.67 vs. 8.05 μmol/L, *p* = 0.045).

## 4. Discussion

The relationship between hypoglycemia and oxidative stress has not been adequately investigated, especially in daily, routine clinical practice. Herein, we show that in persons with diabetes, who experienced spontaneous hypoglycemia in the outpatient diabetes clinics of two major university hospitals, as well as in non-diabetic individuals with a history of hypoglycemic episodes which were evaluated with a prolonged fasting test, there was no difference in the examined OS and antioxidant capacity biomarkers. Only levels of TBARS showed significantly lower values during hypoglycemia as compared with euglycemia (*p* = 0.005).

Thiobarbituric acid reactive substances (TBARS) are formed as a byproduct of lipid peroxidation. The assay uses thiobarbituric acid (TBA) as a reagent and measures malondialdehyde (MDA), a reactive aldehyde produced by lipid peroxidation of polyunsaturated fatty acids [21]. Of note, though, MDA was also measured spectrophotometrically in our study population and did not show any changes. To our knowledge, this is the first study that indicates a possible reduction in biomarkers of lipid peroxidation based on TBARS at the time of hypoglycemia encountered in daily clinical practice.

There is scarce evidence in the literature that TBARS is positively associated with blood glucose levels, albeit more so with hyper- or normo-glycemic and not specifically with hypo-glycemic levels. In a study where TBARS were measured in the serum of subjects with T2D, impaired glucose tolerance, and normal glucose tolerance, TBARS levels were significantly higher in patients with T2D and there was also a positive correlation between TBARS levels and HbA1c, implying that TBARS levels decrease when HbA1c values are low [22]. Furthermore, after 10 days of a very low-calorie diet (around 250 kcal/day), it was shown that TBARS levels decreased and total antioxidant capacity (TAC) increased in plasma, and there was a significant association between TBARS and glucose serum levels [23]. This suggests that TBARS production is reduced during low glucose caused by fasting. However, it should be emphasized that the reduction of glucose did not reach hypoglycemic levels in the specific study (plasma glucose decreased from 93.7 mg/dL [5.2 mmol/L] pre-fasting to 80.4 mg/dL [4.46 mmol/L] post-fasting).

It is well established that hyperglycemia induces an increase in pro-inflammatory cytokines and oxidative biomarkers [6]. It has also been reported that combined treatment with metformin and chitosan-stabilized selenium nanoparticles, which serve as antioxidants, could improve diabetic complications by reducing oxidative stress and restoring glucose homeostasis in high-fat diet feed with low-dose streptozotocin (HFD-STZ) diabetic rats [24,25,26]. On the other hand, hypoglycemia is known to induce an increase in pro-inflammatory mediators, but its relationship with OS is not well elucidated. In hypoglycemic clamp studies [12,13], with blood glucose levels maintained at 2.5 and 2.9 mmol/L (45 and 52.2 mg/dL), a longer duration of hypoglycemia (120 vs. 60 min) resulted in a more remarkable increase in pro-inflammatory mediators. On the other hand, hypoglycemia induced by insulin infusion in non-diabetic male subjects was associated with a rise in pro-inflammatory cytokines, markers of lipid peroxidation by thiobarbituric acid assay, and reactive oxygen species [11]. The increases in oxidative stress and lipid peroxidation markers occurred quickly (at 45 min post insulin injection). They had returned to baseline values at 240 min, possibly associated with the rapid reduction of blood glucose concentrations (to 2.12 mmol/L [38.2 mg/dL] at 30 min post-injection), resulting in a quick release of catecholamines and the stimulation of the inflammatory response. In addition, the baseline TBARS levels in these healthy, non-diabetic people (0.6 μmol/L) were much lower than the euglycemic TBARS levels in our study (9.23 μmol/L), which is consistent with reports in the literature that TBARS levels in people with diabetes are higher than in non-diabetic controls [27]. Furthermore, in another study where 2 h hyperglycemic and hypoglycemic clamps were performed, it was found that hypoglycemia significantly increased markers of oxidative stress and inflammation [28].

In the present study, simple correlation analyses showed a negative correlation between TBARS levels during euglycemia and the age of the participants (r = −0.364, *p* = 0.044). They were significantly higher in males (*p* = 0.045) compared with female subjects, corroborating findings in the literature [29], where TΒARS levels and other oxidative stress biomarkers were higher in healthy young men than in age-matched women. The increased production of ROS in men, as compared with female subjects, may partially explain their greater susceptibility to atherosclerotic cardiovascular events.

Limitations of our study include (a) the relatively small sample size; (b) the fact that the duration and the speed of development of hypoglycemia were unknown but probably very recent and short-lived (blood was drawn immediately when patients reported symptoms consistent with hypoglycemia at the outpatient departments), and thus the levels of the oxidative stress and antioxidant biomarkers may not have had the time to change. No follow-up blood levels were available in the hours after the hypoglycemic episode. Furthermore, other known or unknown confounding factors that may have influenced the levels of the OS and/or other antioxidant parameters (for example, uric acid levels that are known to affect TAC levels [30]) were not measured; (c) although TBARS and MDA are both indicators of lipid peroxidation, the TBARS assay is not as specific as the MDA, since a variety of other compounds of fatty peroxide-derived decomposition products, such as oxidized lipids, saturated and unsaturated aldehydes, sucrose, and urea, interfere with the TBARS assay [21]. Thus, it is generally considered better, albeit more cumbersome, to measure MDA. Our study observed a non-significant increase in ΜDA levels, whereas TBARS levels decreased significantly during hypoglycemia. Consequently, this reduction of TBARS should be viewed with caution since it could be a matter of chance or influenced by unknown confounders [31].

Furthermore, the results of the present study do not advocate using any antioxidants routinely in persons with diabetes, which have not anyway been shown to offer any benefit in routine clinical use, other than in persons with certain deficiencies (for example, specific vitamin or mineral deficiencies, etc.) [32].

## 5. Conclusions

In conclusion, we observed that hypoglycemia encountered in daily, routine clinical practice was not associated with changes in OS or antioxidant biomarkers, except for a reduction in TBARS levels, reflecting possible decreased lipid peroxidation. Since all other redox biomarkers did not show significant changes during hypoglycemia as compared with euglycemia, especially since MDA, another, probably more reliable marker of lipid peroxidation, was not affected, this finding of TBARS decrease is intriguing and should be considered preliminary, needing further investigation in future studies.

## Figures and Tables

**Figure 1 healthcare-10-01509-f001:**
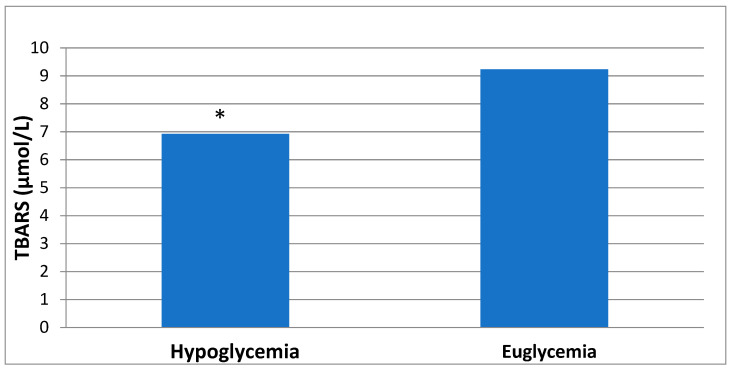
Comparison of TBARS levels between hypoglycemia and euglycemia. * *p* = 0.005 hypoglycemia vs. euglycemia.

**Table 1 healthcare-10-01509-t001:** Demographic, clinical, and laboratory characteristics of participants (mean ± SD).

Variable	T1D	T2D	ND	Total
Number	12	14	5	31
Gender (male) [n (%)]	4 (28.6)	8 (57.1)	2 (14.3)	14 (45.2)
Age (years)	44.3 (17.8)	60.8 (21.3)	47.2 (22.2)	52.2 (21.1)
Weight (kg)	69.0 (15.0)	81.6 (14.1)	69.4 (4.8)	74.7 (14.5)
BMI (kg/m²)	25.1 (4.5)	29.4 (6.1)	24.8 (0.9)	27.0 (5.4)
HbA1c (%)	7.1 (1.1)	7.9 (1.2)	-	7.5 (1.22)
DM duration (years)	21.8 (13.9)	18.3 (9.0)	-	19.9 (11.4)
Glucose Hypo (mmol/L)	3.12 (0.35)	2.37 (0.71)	2.99 (0.31)	2.76 (0.64)
Glucose Eugl (mmol/L)	5.98 (0.82)	6.06 (0.89)	5.47 (1.09)	5.93 (0.89)
eGFR (mL/min/1.73 m²)	90.4 (36.2)	86.9 (44.2)	103.0 (16.8)	90.9 (37.4)

T1D—Type 1 Diabetes Mellitus, T2D—Type 2 Diabetes Mellitus, ND—Non-Diabetic, BMI—Body mass index, Hypo—Hypoglycemia, Eugl—Euglycemia, eGFR—estimated Glomerular Filtration Rate.

**Table 2 healthcare-10-01509-t002:** Comparison of oxidative stress biomarkers and antioxidant capacity indices between hypoglycemia and euglycemia [mean ± SD for normally distributed variables or median (IQR) for non-normally distributed variables].

Variable	Hypoglycemia	Euglycemia	*p* *
ADMA (ng/mL)	40.41 (17.09)	44.62 (19.11)	NS
MDA (μΜ)	5.11 (3.87–6.62)	4.78 (4.11–5.58)	NS
4-HΝΕ (μΜ)	0.78 (0.58–1.11)	0.91 (0.71–1.48)	NS
Protein carbonyls (nmol/L)	12.87 (11.12–18.38)	12.12 (10.14–18.38)	NS
Ox-LDL (ug/mL)	3.05 (0.9)	2.75 (0.85)	ΝS
3-NT (ng/mL)	66.14 (30.64)	68.97 (30.46)	NS
TBARS (μmol/L)	6.55 (4.91–8.92)	9.23 (6.21–11.72)	**0.005**
ABTS (mmol/ABTS/L)	20.72 (5.5)	21.32 (5.75)	NS
Reducing power (μmol/mL)	1.11 (1.18)	1.13 (1.17)	NS
TAC (mmol DPPH/L)	0.86 (0.1)	0.87 (0.1)	NS
Superoxide scavenging capacity (mmol NBT/L)	1.08 (0.27)	1.01 (0.27)	NS
Hydroxyl-radical scavenging capacity (mmol Deoxyribose/mL)	0.02 (0.005)	0.01 (0.001)	NS

ADMA—asymmetric dimethylarginine, MDA—Malondialdehyde, 4-HNE—4-Hydroxynonenal, 3-NT—3-Nitrotyrosine, ox-LDL—Oxidized LDL, TBARS—Thiobarbituric Acid Reactive Substances, ABTS—2,2′-azino-bis(3-ethylbenzothiazoline-6-sulfonate) radical cation, TAC—Total Antioxidant Capacity, NS—Non-significant, * *p* = Comparison among hypoglycemia and euglycemia by paired sample *t*-test or the Wilcoxon signed-ranks test for non-parametric variables.

## Data Availability

Raw data for this study have been deposited in the National and Kapodistrian University of Athens “Pergamos” Repository and can be found at: https://pergamos.lib.uoa.gr/uoa/dl/object/2974878 (accessed on 27 February 2022).

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
