# Peer review of "Association of Hypoglycemia with Biomarkers of Oxidative Stress and Antioxidants: An Observational Study"

_healthcare, 2022, doi:10.3390/healthcare10081509_

Round 1

Reviewer 1 Report

Comments to the author:

The manuscript entitled “ Association of hypoglycemia with biomarkers of oxidative 2 stress and antioxidants. An observational study “ I found the the manuscript is very limited in data presented and also not Novel , these are very urgent and Major concerns, you should  have elucidated several other points and also the accuracy of the detected tests for liver and kidney functions are not presented. I have problems with obtaining just samples from patients and have sharp conclusions for the states.  

I have some concerns which could be valuable for improving the manuscript :

1-     In the abstract “Many sentences were very redundant and very long  “ from line 24-28” and the following sentence was the same ,also contain grammatical and typos mistakes “

2-     The information presented in line 32 was not clear enough and seems to be non-homogenous, this part needs to be presented clearly.

3-     Line 61 : “ All patients with diabetes who developed hypoglycemia were being treated with insulin” this needs clarification , why TYP 2 diabetes was treated with insuline here? Is this the clinical trial or what ?

4-     What are the symptoms of hypoglycemia you based on and why ? why didn’t you based on the level of blood glucose? Revise this in the methods section? Clarify the causes on which you select your patients and why with possible citations?

5-     What are the rationale for the detectable biomarkers ? Why didn’t you estimate the levels of other antioxidants  like GSH ,GPX and CAT , also the levels of NO and PC to be other indicative of OS

6-     What are the detected markers regarding “ acute kidney injury, infection/sepsis, acute liver injury, recent” where are the results?

7-     You mentioned that TBRAS is scarcly associated with hypoglycemia “ I think this needs revision “ find these literatures “: https://pubmed.ncbi.nlm.nih.gov/28216053/, https://pubmed.ncbi.nlm.nih.gov/7733126/”

8-     Line 188 “ “ Hypoglycemia is known to induce an increase in pro-inflammatory mediators, but 188 its relationship with OS is not well elucidated” I think this also needs revision , all these points are mentioned and accurate several times in many literatures.

9-     The figure is very bad quality and presentation.

10-  Discussion needs more attributions and citations for the previous agreements and non.

11-  The Meaning of Post-translational modification is not clear , we should add examples

12-  The first three sentences in the introduction have no reference, although it should have many , for this part.

13-  Also in the section “ 2. ALDH2 and its association with oxidative stress related to aldehyde metabolism “  the readers obtained very crucial information with no references from the line 63-67, you can add the appropriate citation because these are scientific facts.

14-  Line 82 “ Michael addition or Schiff base: should be defined what they mean for better readers

15-  From the line 96 : should add a separate suitable title to facilitate the readers reach his goal with in the review.

16-  From line 103-107, a one sentence which is really too long and also without reference.

17-  Figure 1. Pathway by which aldehyde dehydrogenase 2 (ALDH2) catalyzes aldehyde metabolism 118 in alcohol : even to replace in the legand to ethanol or change in the diagram “ use the same term”

18-  Line 125 “ “ Clinically, ALDH2 deficiency leads to a variety of human diseas” please illustrate more details the sentence is too short.  

19-  The meaning of  “ aldehyde toxicity–related diseases” should be illustrated , not to be confused if aldehydes are beneficial or toxic.

20-  ALDH2, can significantly reverse the increases in alanine aminotransferase 392 (ALT) and aspartate aminotransferase (AST) induced by liver I/R injury in rats, Cite appropriate citations about this line of previously published work.

21-  I found it is better if you could concentrate on the excessive ethanol abuse and chronic effects in the conclusion section.

Reviewer 2 Report

It's interesting to see this study on hypoglycemia. As a result of hypoglycemia, the antioxidant system is down regulated which leads to ROS generation. High levels of free radicals in turn activate the processes leading to DNA destruction and so on.

My queries and suggestions for authors are below:

1. Line No: 50 and 51 -The following words "That effect" in these lines, seem unclear: "the possible link of hypoglycemia with OS has been investigated to that effect"

2. Could you briefly share the reason for omitting biomarkers like Myloperoxidase (MPO), GPX, and hs-CRP, antioxidants such as catalase and glutathione (GSH) reduced in this work.

3. Line No: 37 - What do you mean by MPA? I think it must be MDA

4. Line NO: 62 and 63 - "Hypoglycemia was considered present when symptoms consistent with hypoglycemia occurred [7]", is it possible for you to briefly mention the symptoms here.

5. Authors would have included the lifestyle and diet regimen (because not consuming enough food can cause hypoglycemia) of participants whether sedentary/ active/ overworking.

6. Whether it's hypoglycemia or hyperglycemia, it's been considered that both of them are associated with oxidative stress, I think it's necessary to study the inflammatory markers such as TNF, interleukins (IL-1, IL-2, IL-6, IL-18, IL-10), soluble receptor for advanced glycation end-products (sRAGE), intercellular adhesion molecule-1 (ICAM-1), monocyte chemoattractant protein-1 (MPC-1), etc. What's your take on this? 

7. I noticed the mention of insulin or sulfonylureas but this study did not include the anti-diabetic treatment of the diabetic participants

Reviewer 3 Report

This paper describes a research on OS in two cohorts of DM patients and a control group of non-DM ones.

The study has the merit to investigate a difficult field to be studied, as  it  is the OS study that requires  a good laboratory  background, as is the case of the authors.

But in my opinion the study has great limitations.

MAJOR CONCERNS:

- The number of patients is too small, - and especially the control group, non DM patients, n=5) in order to support the conclusions. To apply a Wilcoxon test and other, you need almost 6-8 individuals.

The eGFR of the patients is normal/high, but if you observe the standard deviation, some of the patients in both groups, DM1 and DM2, had an eGFR below 60 mL/min/1,73m2. This fact means that some patients had a renal insufficiency (RI) and, in this situation , the study of OS may be affected by such condition.

- On the other hand, you do not adequately describe the concomitant treatment, - besides insulin-,   speciacially of Type 2 DM patients. Were some of them under sulphonyl ureas  treatment ?. This is a formal contraindication for this treatment in the presence of an eGFR below 60 mL/min/1,73 m2. 

- The values of TBATS may be expressed by separate in type 1 and type 2 DM patients , in whom the OS  study may be quite different.

- - The conditions of the extraction and transport to the laboratory of the blood samples may be important when study OS. This aspect is not clearly described in M & M.

- MINOR final comment.

Which are the practical applications of the results of this study to the daily practice ?. Because as you perfectly know, the study of OS is complex and it is  not cheap. Would you mean that all type 1 and type 2 DM paties may be systematically studied for OS situation and all of them may  received antioxidants ? 

Round 2

Reviewer 1 Report

The manuscript entitled “Association of hypoglycemia with biomarkers of oxidative 2 stress and antioxidants. An observational study “I still find the manuscript is very limited in data presented and also not Novel , the number of studied cases are very limited to judge the states.

There are some papers should be cited in the discussion section about the correlation between the OS and diabetes and the antioxidants that can cooperate to combat the OS:

1.      Stabilized-chitosan selenium nanoparticles efficiently reduce renal tissue injury and regulate the expression pattern of aldose reductase in the diabetic-nephropathy rat model.

2.      Chitosan-stabilized selenium nanoparticles alleviate cardio-hepatic damage in type 2 diabetes mellitus model via regulation of caspase, Bax/Bcl-2, and Fas/FasL-pathway.

3.      Chitosan-Stabilized Selenium Nanoparticles and Metformin Synergistically Rescue Testicular Oxidative Damage and Steroidogenesis-Related Genes Dysregulation in High-Fat Diet/Streptozotocin-Induced Diabetic Rats

Reviewer 3 Report

The authors have satisfactory answered all my suggestions. In my opinion the reply to question number 6 may be added at the end of the discussion in order to clarify the conclusion.
